# Ameliorative Effect of Coenzyme Q10 on Phenotypic Transformation in Human Smooth Muscle Cells with *FBN1* Knockdown

**DOI:** 10.3390/ijms25052662

**Published:** 2024-02-25

**Authors:** Xu Zhang, Zhengyang Zhang, Sitong Wan, Jingyi Qi, Yanling Hao, Peng An, Yongting Luo, Junjie Luo

**Affiliations:** Department of Nutrition and Health, China Agricultural University, Beijing 100193, China; zhangx94@cau.edu.cn (X.Z.); zzy2580a@cau.edu.cn (Z.Z.); adeline7wan@cau.edu.cn (S.W.); zb20213311055@cau.edu.cn (J.Q.); haoyl@cau.edu.cn (Y.H.); an-peng@cau.edu.cn (P.A.)

**Keywords:** Coenzyme Q10, Marfan syndrome, mitochondrial dysfunction, smooth muscle cells, phenotypic transformation, the double-hit theory

## Abstract

Mutations of the *FBN1* gene lead to Marfan syndrome (MFS), which is an autosomal dominant connective tissue disorder featured by thoracic aortic aneurysm risk. There is currently no effective treatment for MFS. Here, we studied the role of mitochondrial dysfunction in the phenotypic transformation of human smooth muscle cells (SMCs) and whether a mitochondrial boosting strategy can be a potential treatment. We knocked down *FBN1* in SMCs to create an MFS cell model and used rotenone to induce mitochondrial dysfunction. Furthermore, we incubated the *shFBN1* SMCs with Coenzyme Q10 (CoQ10) to assess whether restoring mitochondrial function can reverse the phenotypic transformation. The results showed that *shFBN1* SMCs had decreased *TFAM* (mitochondrial transcription factor A), mtDNA levels and mitochondrial mass, lost their contractile capacity and had increased synthetic phenotype markers. Inhibiting the mitochondrial function of SMCs can decrease the expression of contractile markers and increase the expression of synthetic genes. Imposing mitochondrial stress causes a double-hit effect on the *TFAM* level, oxidative phosphorylation and phenotypic transformation of *FBN1*-knockdown SMCs while restoring mitochondrial metabolism with CoQ10 can rapidly reverse the synthetic phenotype. Our results suggest that mitochondria function is a potential therapeutic target for the phenotypic transformation of SMCs in MFS.

## 1. Introduction

Marfan syndrome (MFS) is an autosomal dominant disease caused by *FBN1* (fibrillin-1) gene mutations, and the morbidity is 1/5000 individuals [1,2]. MFS patients suffer from abnormal connective tissue and present extended bones, lens dislocation and reduced life expectancy primarily attributable to aortic aneurysms [3]. The vascular remodeling of smooth muscle cells (SMCs) is a key event in the occurrence and development of aortic aneurysms [4]. SMCs are mainly in the contractile phenotype and are responsible for the contraction and relaxation of the blood vessel [5]. In response to injury, aging or diseases, SMCs change their metabolic features and function toward synthetic phenotypes. Synthetic SMCs are characterized by proliferation, hypertrophy, migration, senescence and an abnormal extracellular matrix, which may damage the integrity of the vascular wall [6]. Mitochondria participate in vascular remodeling through multiple mechanisms [7]. For instance, mitochondrial fusion and fission control the proliferation, migration and phenotypic transformation of SMCs [8]. Abnormal mitophagy can accelerate the senescence and apoptosis of SMCs [9]. Mitochondrial DNA (mtDNA) damage causes defects in oxidative phosphorylation and directly promotes atherosclerosis in a mouse model [10]. A mouse model with mitochondrial dysfunction, specifically in vascular SMCs (*Myh11-Cre^ERT2^Tfam^−/−^* mice), showed synthetic phenotype and aortic remodeling [1]. Otherwise, vascular SMCs are highly energy-consuming and rely on the ATP and metabolites from mitochondria.

Mitochondria are essential organelles with a plethora of functions. Besides energy production, mitochondria are the hub of the tricarboxylic acid (TCA) cycle and contribute to the metabolism of amino acids, lipids, nucleotides and ions, the biosynthesis of iron–sulfur (Fe-S) clusters, cellular homeostasis and immune and inflammatory response [11]. Mitochondria have been recognized as pathogenesis or modification factors in many cardiovascular pathologies [11], including Stanford type A aortic dissection (STAAD/AD) [12], atherosclerosis [13] and heart failure [14]. In MFS patients and *Fbn1^C1039G/+^* mice, RNA-sequencing analysis indicates a declined mitochondrial function and mitochondrial biogenesis. Nicotinamide riboside (NR) treatment can increase *Tfam* levels and mitochondrial respiration in MFS mice primary vascular SMCs and the primary dermal fibroblasts of MFS patients [1]. In a mouse model of Marfan syndrome, NR treatment can revert the remodeling of the aortic wall, aortic dilation and medial degeneration [1]. However, whether mitochondrial dysfunction exacerbates the phenotypic transformation of SMCs caused by *FBN1* deficiency and the beneficial effect of other supplements targeting the mitochondria of SMCs have not been assessed.

In this study, we investigate the effect of mitochondrial function on the phenotypic transformation of SMCs in the context of *FBN1* knockdown with short hairpin RNA targeting FBN1 (*shFBN1*), an MFS cell model. Our results indicate that mitochondrial dysfunction causes a double hit on *shFBN1* SMCs, and targeting mitochondria by Coenzyme Q10 (CoQ10) can alleviate or reverse the vascular remodeling.

## 2. Results

### 2.1. SMCs with FBN1 Knockdown Present Features of Decreased TFAM Expression, mtDNA Content and Changed Phenotypic Transformation

To investigate the *TFAM* expression, mtDNA content and phenotypic transformation marker gene levels in the MFS cell model, we induced *FBN1* knockdown by lentivirus-mediated shRNA transfection (Figure 1A). *FBN1* knockdown efficiency was confirmed by analyzing the *FBN1* mRNA level (Figure 1B). The *TFAM* expression level significantly decreased after 72 h of lentiviral transfection (Figure 1C). In accordance with *TFAM’s* role in regulating mtDNA copy number [15], the mtDNA level in SMCs decreased significantly (Figure 1D). In addition, the mitochondrial mass is decreased after *FBN1* knockdown (Appendix A). Figure 1E presents the gene expression level of *ACTA2*, *CNN1*, *TAGLN*, *MYL9* and *MYH11*. These genes are involved in contractile phenotype and significantly decreased after *FBN1* knockdown. The gene expression of synthetic phenotype markers *COL3A1*, *COL1A2*, *FN1*, *CXCL2* and *IL-6* are significantly up-regulated (Figure 1F). The most significant change genes are inflammatory cytokines *CXCL2* and *IL-6*, which increase about 15.1 and 4.4 times, respectively. These results show that *FBN1* deficiency causes lower *TFAM* expression, mtDNA content, mitochondrial mass and changed phenotypic transformation of SMCs.

### 2.2. Rotenone-Induced Mitochondrial Dysfunction Elicits the Phenotypic Transformation of SMCs

Rotenone is a mitochondrial electron transport chain complex I inhibitor, which can promote reactive oxygen species (ROS) production and induce mitochondrial dysfunctions [16]. To assess whether the mitochondrial respiratory dysfunction affects SMCs’ phenotype, we measured the mtDNA copy number and the expression of phenotypic transformation marker genes in SMCs with rotenone treatment. In Appendix A, when the dose of rotenone is 500 nM, the SMCs showed significantly lower levels of mtDNA compared with the control group after 12 h. When treatment time was prolonged to 24 h, the mtDNA copy number decreased by two-thirds. The expression of contractile marker *ACTA2* was significantly decreased after 12 h, while *TAGLN* and *MYH11* presented a significant decline after 24 h (Appendix A). Rotenone induced up-regulation of synthetic genes *COL3A1*, *FN1* and *CXCL2* after 12 h (Appendix A). The fold change of chemokine *CXCL2* gene expression reaches 6 after 24 h treatment. Moreover, the decrease in the mtDNA copy number and phenotypic transformation is aggravated under higher rotenone doses (1 μM). The mtDNA copy number significantly declines after 12 h rotenone treatment (Figure 2A,B). The expression of the above synthetic genes was up-regulated, and contractile marker expression was down-regulated even after 12 h (Figure 2C–H). These data indicate that rotenone can induce an mtDNA copy number decline and the phenotypic switch of SMCs in a time- and dose-dependent manner.

### 2.3. Mitochondrial Dysfunction Aggravates Synthetic Phenotype in an SMC Model of MFS

To investigate whether there is a double-hit effect of mitochondrial decline on the phenotypic transformation of SMCs in the context of *FBN1* deficiency, we induced further mitochondrial dysfunction in *shFBN1* SMCs with rotenone (Figure 3A). After 24 h treatment, rotenone decreased the expression of *TFAM* (Figure 3B). The intracellular ATP (Figure 3C) was decreased, and the extracellular L-Lactate level (Figure 3D), as an indicator of glycolysis, was higher than the *shControl* and *shFBN1* groups. The rotenone treatment significantly decreased the expression of the contractile phenotype markers *ACTA2*, *TAGLN* and *MYH11* in *shFBN1* SMCs (Figure 4A). The gene expression of *CNN1* and *MYL9* in the combined group had a lower transcriptional level compared to the *shFBN1* SMC group. In Figure 4B, rotenone elevates the gene expression levels of *COL3A1*, *COL1A2*, *FN1*, *CXCL2* and *IL-6*, especially *COL3A1*, *FN1* and *IL-6*. The representative immunoblotting of MYH11, α-SMA, CNN1, COL1A2, IL-6 and CXCL2 is shown in Figure 4C. These data support that SMCs under the hits of *FBN1* deficiency and mitochondrial dysfunction have worse mitochondrial respiration, rewire the metabolism toward glycolysis and display severe synthetic and inflammatory characteristics.

### 2.4. CoQ10 Reverts Phenotypic Transformation in a Cell Model of MFS

Coenzyme Q10 (CoQ10) is an antioxidant and a central component in the mitochondrial electron transport chain (ETC). CoQ10 supplementation has been proposed as a strategy to promote mitochondrial function and relieve inflammation [11]. To investigate the therapeutic potential of CoQ10 in SMCs of synthetic phenotype, *shFBN1* SMCs were treated with CoQ10. The exposure of *shFBN1* SMCs to CoQ10 for 24 h increased the gene expression of *TFAM* (Figure 5B) and intracellular ATP (Figure 5C) and decreased extracellular L-Lactate (Figure 5D) levels. Furthermore, CoQ10 increased the expression of the contractile phenotype markers *ACTA2*, *CNN1*, *TAGLN*, *MYL9* and *MYH11* (Figure 6A) and decreased the synthetic phenotype markers *COL3A1*, *COL1A2*, *FN1*, *CXCL2* and *IL-6* to the control group levels (Figure 6B). The representative immunoblotting of MYH11, α-SMA, CNN1, COL1A2, IL-6 and CXCL2 is shown in Figure 6C. Taken together, our data indicate that the CoQ10 supplementation restores *TFAM* levels and improves mitochondrial respiration in SMCs with *FBN1* deficiency.

## 3. Discussion

We found that the knockdown of *FBN1* leads to a deceased TFAM expression, a decline in the mtDNA copy number and a deceased mitochondrial mass. This is in accordance with the reported reduced mitochondrial biogenesis and activity of function regulators in *Fbn1*^C1039G/+^ mice, a mouse model of MFS [1]. We also identified a significant increase in inflammatory cytokines *CXCL2* and *IL-6.* Inflammation is observed in the aortic tissue of Marfan patients and a mouse model [17,18]. *TFAM* increase induces mtDNA leakage from mitochondria into the cytoplasm and activates the cGAS-STING pathway [19], which may explain the reason why abatacept (a T-cell-specific inhibitor) failed to slow the aortic dilatation rate in a Marfan mouse model [17]. In addition, mitochondrial dysfunction can induce the phenotype switch of human SMCs. After exposure to the mitochondrial respiratory complex Ⅰ inhibitor rotenone, the expression of synthetic genes was up-regulated, while the contractile marker expression was down-regulated according to the exposure time and dose. Compared to previous research, we confirm a double-hit effect of mitochondrial dysfunction and *FBN1* knockdown on SMCs. SMCs carrying MFS mutations present reduced expression of all mitochondrial complex subunits and rewire the metabolism toward glycolysis [1]. Mitochondrial complex Ⅰ inhibitor rotenone further reduced the activity of mitochondrial biogenesis *TFAM* and mitochondrial respiration and boosted the phenotypic transformation of SMCs. Although 1 μM rotenone had no significant effect on *TFAM* levels in the *shControl* SMCs after 24 h treatment (Appendix A), it induced an ATP level decline (Appendix A). Rotenone can also induce mitochondrial ROS production which may aggravate the inflammation and senescence features of SMCs with *FBN1* deficiency [1,16].

For the medications of MFS patients, the 2010 AHA/ACC Thoracic Aortic Disease guidelines recommended β-adrenergic receptor blockers (β-blockers) for MFS patients with aortic aneurysms to slow aortic dilatation [20]. β-blockers can reduce heart rate and blood pressure, thus decreasing the force of ejected blood on the aorta. Angiotensin II receptor blocker (ARB), losartan, was found to prevent aneurysm growth in an MFS mouse model [21]. However, the results of clinical trials are mixed [22,23,24,25]. In a double-blind 5-year study, irbesartan, an ARB, significantly lowered the rate of change in absolute root diameter [22]. A randomized controlled trial demonstrated that losartan can reduce aortic dilatation in adults with MFS [23], whereas a 3-year double-blind trial demonstrated that losartan failed to decrease aortic dilatation [26]. When the aorta diameter reaches 5.0 cm in adults, a timely surgical repair of the aorta is still needed [27]. In addition, there is currently no medication for mitochondrial dysfunctions in MFS. Except for secondary mitochondrial decline induced by nuclear genes (for example, *FBN1*), mtDNA mutations are important inducements of mitochondrial dysfunction. Mitochondria contain their genetic system. mtDNA encodes critical proteins of OXPHOS (mitochondrial oxidative phosphorylation system), the mitochondrial ribosome and tRNA [28]. Severely deleterious mtDNA mutations can cause significant mitochondrial gene dysfunctions and physiological changes. The mutations in nuclear genes can be modified factors to aggravate the symptoms of mtDNA mutation-related diseases [29]. For example, mt.A1555G mutation is related to aminoglycoside-induced and nonsyndromic deafness, but it is insufficient to induce a deafness phenotype solely. The mutation of the nuclear gene TRMU (tRNA 5-methylaminomethyl-2-thiouridylate methyltransferase) leads to failure in mitochondrial tRNA metabolisms. Abnormal mitochondrial tRNA aggravates the mitochondrial dysfunction related to the mt.A1555G mutation, exceeding the threshold and manifesting a deafness phenotype [30]. It can be speculated that an MFS patient with mitochondrial dysfunction will have a more severe OXPHOS disorder and inflammation.

Our results indicated that CoQ10 treatment can improve the contractile functions of SMCs with *FBN1* dysfunction. CoQ10 has antioxidant and anti-inflammatory abilities and can directly reduce ROS, and its fat solubility makes it easier to cross cell membranes [31]. Oxidative stress is detected in the plasma MFS patients and MFS mice mode [32,33], involving decreased levels of superoxide dismutase (SOD) and enhanced levels of NAD(P)H oxidase. NR treatment decreased the gene expression of matrix metalloproteinase (MMP) enzymes and synthetic phenotype markers in murine and human MFS cells. The beneficial effect was attributed to an increase in mitochondrial biogenesis by NR [1]. These results suggest that small molecules with antioxidant, anti-inflammation and increased mitochondrial biogenesis functions, for example, pyrroloquinoline quinone (PQQ) [17], may be beneficial for the treatment of Marfan syndrome. The corresponding data on the effects of *shFBN1*, rotenone and CoQ10 on *TFAM* and phenotypic transformation markers of SMCs are summarized in Appendix A for a better comparison. Considering the different protective mechanisms of NR, CoQ10 or other mitochondria-targeting molecules, those with the functions of improving *TFAM* levels or combined treatments may have better therapeutic effects.

Currently, symptomatic management for patients with these complex symptoms is a common treatment. However, the solution for the root cause is correcting the mitochondrial gene abnormal expression or mtDNA mutation that causes the biochemical defect. Mitochondrial DNA of Marfan patients can be sequenced to analyze the pathogenic mutation. mtDNA mutations can be detected by targeted mtDNA sequencing or filtered from other omics data [34]. The elimination of mutant mtDNA technologies has made giant leaps in past decades. Zinc-finger nucleases (ZFNs) and transcription activator-like effector nucleases (TALENs) are protein-based editing tools and can be guided into mitochondria by localization signals. mitoZFNs and mitoTALENs have been designed to manipulate heteroplasmy in vitro or in vivo [35,36]. However, reducing mutant mtDNA load is ineffective in treating homozygous or single-base mutations. Mutations in mtDNA are associated with many human diseases, and about 95% of them are caused by single-base mutations that could potentially be corrected by base editing [37]. Currently, deaminas and TALEN-based editors can achieve a C-to-T or A-to-G base mutation of the mitochondrial genome [38,39]. The tools are continually developed to improve editing efficiency and accuracy [40]. Along with traditional mitochondria-targeting antioxidants, the mitochondrial editing technology is prospected to open new possibilities in manipulating diseases with mitochondrial dysfunctions.

## 4. Materials and Methods

### 4.1. Cell Procedures

Human aortic smooth muscle cells (SMCs) were purchased (CP-H081, Procell, Wuhan, China) and cultured in the Smooth Muscle Cell Medium (1101, ScienCell, Carlsbad, CA, USA) supplemented with FBS (10099141, Gibco, Carlsbad, CA, USA) and 10% smooth muscle cell growth supplement (1162, ScienCell, Carlsbad, CA, USA). Cells were maintained in 5% CO_2_ at 37 °C. For stimulation of SMCs, cells were treated with 500 nM/1 μM rotenone (HY-B1756, MedChemExpress, Monmouth Junction, NJ, USA) or 80 μM CoQ10 (C805246, Macklin, Shanghai, China) for the number of hours indicated in the figure. 

### 4.2. Lentivirus Production and Infection

Lentiviruses expressing short hairpin RNA (shRNA) were used to knock down *FBN1* in human SMCs. The lentiviruses containing control shRNA and *shFBN1* (sequence in Appendix A) were purchased from GenePharma (Shanghai, China). The SMCs were transducted by lenti-*shFBN1* (MOI = 10) with 8 μg/mL polybrene (H8761, Solarbio, Beijing, China). After 72 h/24 h of culture (as indicated in the figures), the cells were harvested or stimulated for further experimentations.

### 4.3. RNA and DNA Extraction

The total RNA of cultured human SMCs was extracted using TRIzol (15596026, Thermo, Carlsbad, CA, USA), according to the manufacturer’s instructions. The total DNA of cultured cells was extracted with a Genomic DNA Purification Kit (B0007, EZBioscience, Roseville, CA, USA).

### 4.4. Real-Time PCR

Qualified RNA (1 μg) of each sample was reverse-transcribed into cDNA with a Reverse Transcription Reagent kit (R323, Vazyme, Nanjing, China), according to the manufacturer’s instructions. Real-time quantitative PCR (qPCR) analysis was performed on the QuantStudio™ 5 Real-Time PCR Systems (A28575, Applied Biosystems, Carlsbad, CA, USA) with SYBR Green PCR mix (Q711, Vazyme, Nanjing, China), according to the following program: pre-denaturation at 95 °C for 30 s. (one cycle), denaturation at 95 °C for 5 s, annealing at 60 °C for 20 s and elongation at 72 °C for 20 s (40 cycles); melt curve analysis was performed with the default program. All samples were amplified using three biological replicates per sample. The QuantStudio Design & Analysis Software (v1.5.2, Applied Biosystems, Carlsbad, CA, USA) was used to obtain CT values. The amount of target mRNA in each sample was estimated with the comparative CT method, using *GAPDH* for normalization. The primer sequences for PCR are summarized in Appendix A.

### 4.5. mtDNA Copy Number Analysis

mtDNA copy number was assessed using real-time PCR. Primers for mtDNA copy number quantification were designed according to a previously described protocol [41] (Appendix A). The human mtDNA primer and human *B2M* (used as a reference gene) primer were used to amplify the respective products from human total DNA.

### 4.6. Mitochondrial Mass

To determine the mitochondrial mass, SMCs were harvested and stained with 100 nM MitoTracker Deep Red FM (M22426, Invitrogen, Carlsbad, CA, USA) at 37 °C for 30 min. The mean fluorescence of samples was quantified in the Allophycocyanine (APC) channel (excitation: 644 nm; emission: 665 nm) for mitochondrial mass on a flow cytometer (Influx, BD, Franklin Lakes, NJ, USA). Alive singlets were gated in the forward scatter (FSC)/side scatter (SSC) plot, and the respective channel’s mean fluorescence was quantified and corrected by the mean fluorescence of an unstained SMC sample.

### 4.7. Intracellular ATP Assay

The treated SMCs were lysed on ice before being centrifuged at 12,000× *g* at 4 °C for 5 minutes, and the supernatant was obtained to detect the intracellular ATP levels. Intracellular ATP levels of SMCs were determined using an Enhanced ATP Assay Kit (S0027, Beyotime, Shanghai, China). The ATP-detecting solution was added to a 96-well plate and incubated at room temperature for 5 min away from light. The lysate of cells was then added to the wells, and the luminescence signals were determined using a microplate reader (Agilent BioTek, Winooski, VT, USA). Total ATP levels were normalized to total protein of cell extracts.

### 4.8. Extracellular L-Lactic Acid Assay

The treated SMCs were harvested and lysed using an ultrasonic cell crusher (power: 300 w, crushing time: 3 s, interval time: 7 s, total working time: 3 min). Extracellular lactate determination was performed with an L-lactic acid (LA) Content Assay Kit (BC2235, Solarbio, Beijing, China). LA generates pyruvate catalyzed by lactate dehydrogenase, while NAD^+^ is reduced to NADH and H^+^, which can transfer to PMSH_2_ to reduce MTT to produce purple material. The absorbance was measured at 570 nm using a microplate reader. Extracellular L-LA levels were normalized to total protein of cell extracts.

### 4.9. Immunoblotting

For Western blot analysis, proteins of SMCs were separated by 8% SDS-polyacrylamide gel electrophoresis and then transferred onto polyvinylidene fluoride (PVDF) membranes of 0.45 μm pore size. The membranes were blocked by 5% non-fat milk in TBS-T (20 mM Tris, 137 mM NaCl and 0.1% Tween-20) for 2 h and then incubated with primary antibodies overnight at 4 °C. Followed by TBS-T washes and incubation with horseradish peroxidase (HRP)-conjugated secondary antibodies (Beyotime, China) for 1 h. The immunoblot signal was visualized using a chemiluminescence reagent (Pierce, USA) and ChemiScope 6100 imaging system (Qinxiang Instruments, Shanghai, China).

The following antibodies were used: MYH11 (ab124679, Abcam, Cambridge, UK), α-SMA (#48938, CST, Danvers, MA, USA), CNN1 (24855-1-AP, Proteintech, Rosemont, IL, USA), COL1A2 (AF7001, Affinity Biosciences, Cincinnati, OH, USA), CXCL2 (PA5-79109, Thermo, Carlsbad, CA, USA), IL-6 (ab259341, Abcam, Cambridge, UK) and β-actin (66009, Proteintech, Rosemont, IL, USA).

### 4.10. Statistics

The results of all experiments are presented as mean ± SD (standard deviation). All experiments were performed with three biological replicates. For comparison between two groups, two-tailed Student’s *t*-test was used for significance testing. For comparisons among three groups, one-way ANOVA was used to evaluate statistical differences. GraphPad Prism software (version 9, San Diego, CA, USA) was used to conduct all the statistical analyses. The statistical significance was indicated as * *p* < 0.05, ** *p* < 0.01 and *** *p* < 0.001.

## 5. Conclusions

In conclusion, the present work highlighted the benefits of mitochondrial antioxidant CoQ10 to the phenotypic transformation of human SMCs subject to *FBN1* knockdown, an MFS cell model. We also identified the double-hit effect of mitochondrial dysfunction and *FBN1* knockdown. These results are summarized in Figure 7 and indicate a need to focus on the mitochondrial health of Marfan patients and propose mitochondria as a potential target for MFS medications.

## Figures and Tables

**Figure 1 ijms-25-02662-f001:**
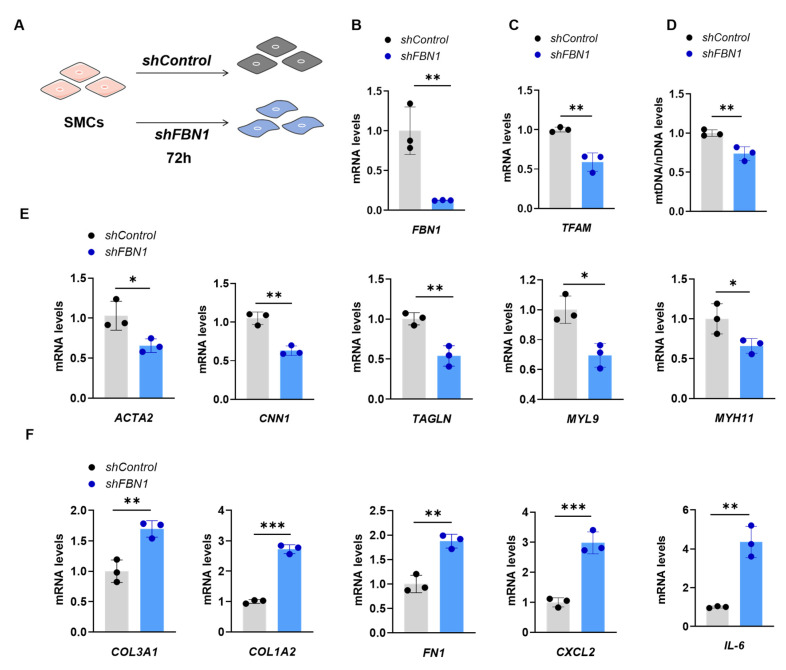
Decreased *TFAM* expression and mtDNA level and changed phenotypic transformation of SMCs with *FBN1* knockdown. (**A**) Schematic of *FBN1* knockdown with lentiviral transduction. (**B**) Knockdown efficiency of *shFBN1*. (**C**) Quantitative PCR analysis of quantitative reverse-transcription PCR analysis of *TFAM* mRNA expression and (**D**) relative mtDNA content. (**E**) Quantitative reverse-transcription PCR analysis of contractile phenotype markers *ACTA2*, *CNN1*, *TAGLN*, *MYL9* and *MYH11*. (**F**) Quantitative reverse-transcription PCR analysis of synthetic phenotype markers *COL3A1*, *COL1A2*, *FN1*, *CXCL2* and *IL-6*. * *p* < 0.05, ** *p* < 0.01, *** *p* < 0.001. The *p* value was calculated using two-tailed Student’s *t*-test. The numbers for the analysis in B-F were 3. SMCs: smooth muscle cells; TFAM: mitochondrial transcription factor A.

**Figure 2 ijms-25-02662-f002:**
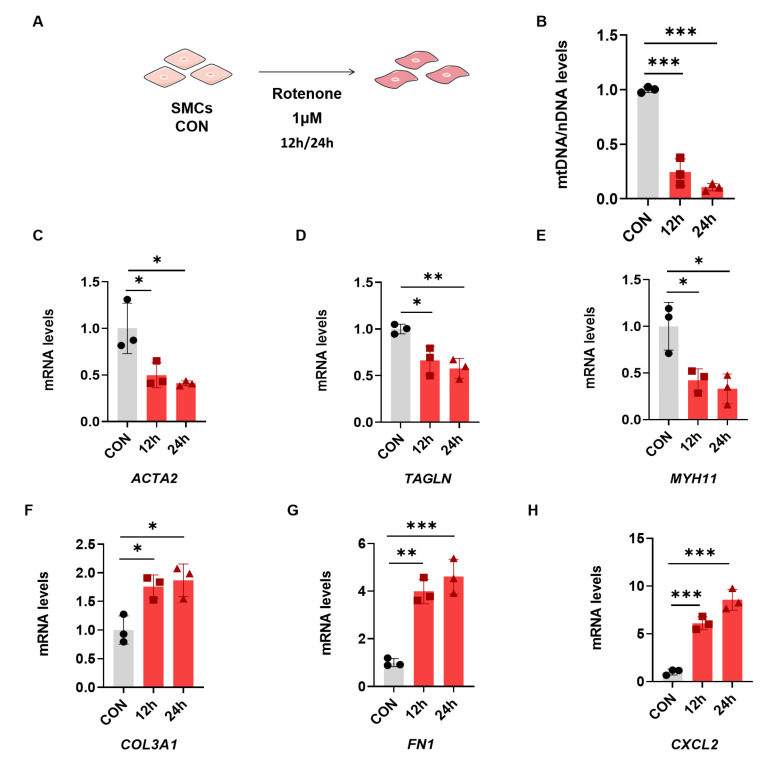
Rotenone can induce the phenotypic switch of SMCs. (**A**) Schematic of rotenone treatment. Quantitative reverse-transcription PCR analysis of mtDNA content (**B**), *ACTA2* (**C**), *TAGLN* (**D**), *MYH11* (**E**), *COL3A1* (**F**), *FN1* (**G**) and *CXCL2* (**H**). * *p* < 0.05, ** *p* < 0.01, *** *p* < 0.001. The *p* value was calculated using one-way ANOVA. The numbers for the analysis in (**B**–**H**) were 3. SMCs: smooth muscle cells.

**Figure 3 ijms-25-02662-f003:**
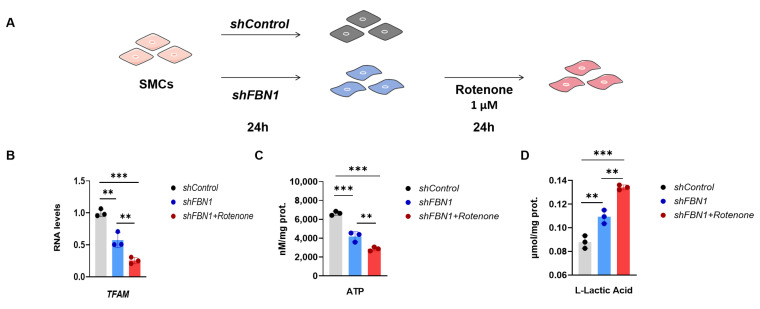
Rotenone aggravates the decrease in *TFAM* level and oxidative phosphorylation of *shFBN1* SMCs. (**A**) Schematic of *FBN1* knockdown and rotenone treatment. (**B**) Quantitative reverse-transcription PCR analysis of *TFAM* mRNA expression. (**C**) The ATP level of SMCs. (**D**) The L-lactic acid level of SMCs. ** *p* < 0.01, *** *p* < 0.001. The *p* value was calculated using one-way ANOVA. The numbers for the analysis in (**B**–**D**) were 3. SMCs: smooth muscle cells; TFAM: mitochondrial transcription factor A.

**Figure 4 ijms-25-02662-f004:**
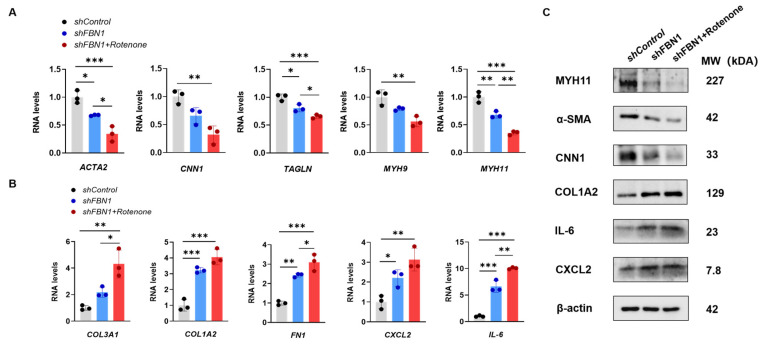
The double-hit effect of mitochondria and *FBN1* dysfunction on the phenotypic transformation of SMCs. (**A**) Quantitative reverse-transcription PCR analysis of contractile phenotype markers *ACTA2*, *CNN1*, *TAGLN*, *MYL9* and *MYH11*. (**B**) Quantitative reverse-transcription PCR analysis of synthetic phenotype markers *COL3A1*, *COL1A2*, *FN1*, *CXCL2* and *IL-6*. (**C**) Representative immunoblot analysis of MYH11, α-SMA, CNN1, COL1A2, IL-6 and CXCL2; β-actin was used as the protein loading control. * *p* < 0.05, ** *p* < 0.01, *** *p* < 0.001. The *p* value was calculated by one-way ANOVA. The numbers for the analysis in (**A**,**B**) were 3.

**Figure 5 ijms-25-02662-f005:**
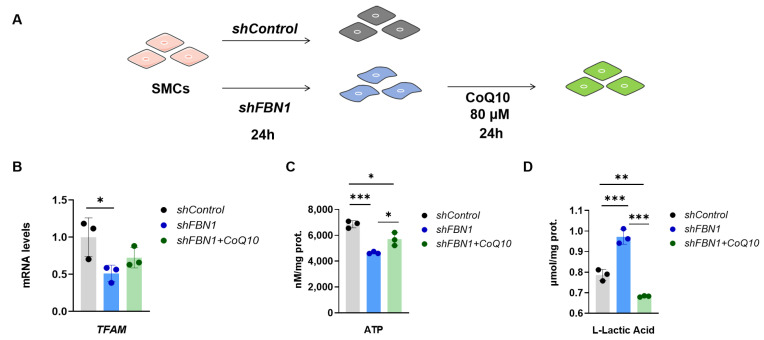
CoQ10 (80 μM) treatment for 24 h can restore the TFAM level and oxidative phosphorylation of SMCs subject to *FBN1* knockdown. (**A**) Schematic of *FBN1* knockdown and CoQ10 treatment. (**B**) Quantitative reverse-transcription PCR analysis of *TFAM* mRNA expression. (**C**) The ATP level of SMCs. (**D**) The L-lactic acid level of SMCs. * *p* < 0.05, ** *p* < 0.01, *** *p* < 0.001. The *p* value was calculated using one-way ANOVA. The numbers for the analysis in (**B**,**C**) were 3. SMCs: smooth muscle cells; CoQ10: Coenzyme Q10.

**Figure 6 ijms-25-02662-f006:**
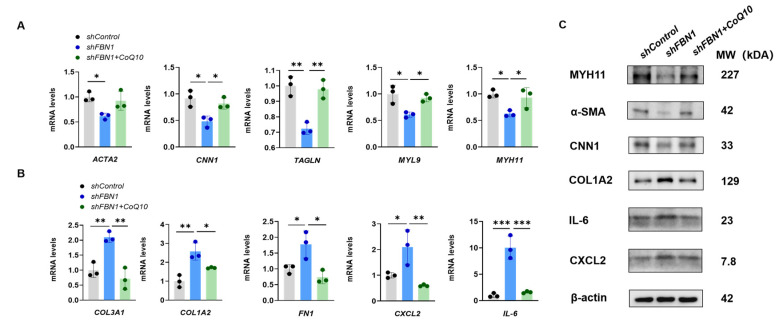
CoQ10 (80 μM) treatment for 24 h can reverse the phenotypic transformation of human smooth muscle cells subject to *FBN1* knockdown. (**A**) Quantitative reverse-transcription PCR analysis of contractile phenotype markers *ACTA2*, *CNN1*, *TAGLN*, *MYL9* and *MYH11*. (**B**) Quantitative reverse-transcription PCR analysis of synthetic phenotype markers *COL3A1*, *COL1A2*, *FN1*, *CXCL2* and *IL-6*. (**C**) Representative immunoblot analysis of MYH11, α-SMA, CNN1, COL1A2, IL-6 and CXCL2; β-actin was used as the protein loading control. * *p* < 0.05, ** *p* < 0.01, *** *p* < 0.001. The *p* value was calculated using one-way ANOVA. The numbers for the analysis in (**A**,**B**) were 3. CoQ10: Coenzyme Q10.

**Figure 7 ijms-25-02662-f007:**
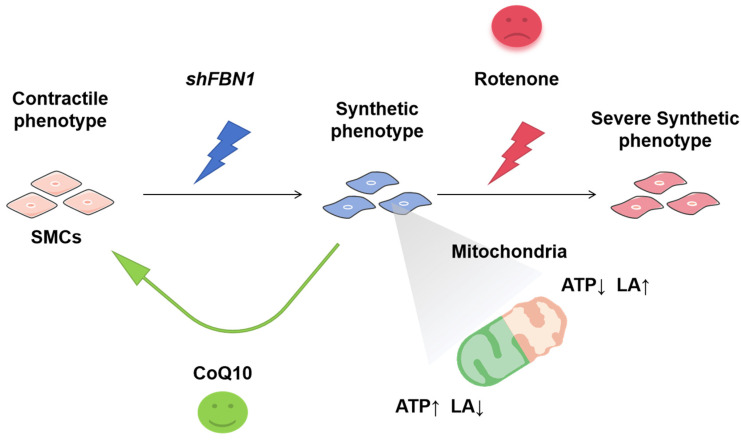
Targeting mitochondria can reverse or aggravate the phenotypic switch of *shFBN1* SMC. Part of the elements of the proposed model are available online: http://smart.servier.com/ (accessed on 4 January 2024). SMCs: smooth muscle cells; CoQ10: Coenzyme Q10; LA: lactic acid.

## Data Availability

Data are contained within the article and Appendix A.

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
