# Peer review of "Ameliorative Effect of Coenzyme Q10 on Phenotypic Transformation in Human Smooth Muscle Cells with FBN1 Knockdown"

_ijms, 2024, doi:10.3390/ijms25052662_

Round 1

Reviewer 1 Report

Comments and Suggestions for Authors

Remarks on the manuscript entitled "Ameliorative Effect of Coenzyme Q10 on the Phenotypic Transformation of Human Smooth Muscle Cells in a Marfan Syndrome Cell Model”.

 In FBN1-deficient human smooth muscle cells, the authors investigated the effect of rotenone and coenzyme Q10 on mitochondrial function. I like the topic of this manuscript, however, in my opinion there are some points that should be improved prior to publication.

Major comment:

Both 4.3 (RNA and DNA Extraction) and 4.4 (Real-Time PCR) need to be revised, as “reverse transcription” is not part of RNA/DNA extraction and the “program” of real-time PCR also includes elongation (primer extension) step. In addition, “5 and 30 sec.“ (line 289) should be specified.

 Minor comments:

- Lines 13 and 62: Explain “sh” in shFBN1.

- Line 40: “mtDNA (mitochondrial DNA)” => “Mitochondrial DNA (mtDNA)”.

- All Figures: The color code is difficult to see (too small) => increase size for color code/circle.

- Figure 1: Specify “D” in the legend.

- Figures 1F, 4B and 6B: I am wondering whether the CXCL2 RNA level in 72h shFBN1 cells is indeed/actually ~7.5 times higher (~15) than in 24h shFBN1 cells (~2). Please double check and revise if needed.

- Figures 1, 4, and 6 as well as Figures 3B and 5B: I am wondering whether the authors could summarize the corresponding data in one (supplementary) figure for better comparison of the effect of shFBN1, rotenone and CoQ10.

- Figure 5: Specify “D” and double check “B” and “C” in the legend.

- Figure 7: Where is this referenced in the text?

- Lines 206, 204-208: Double check the capitalization of irbesartan and losartan and use it consistently.

- Line 214: Specify OXPHOS (mitochondrial oxidative phosphorylation system).

- Line 309: Double check “g” in “12,000g”.

- Line 317: Revise “3min”.

- Revise “knock-down” vs “knockdown” and use it consistently in the entire manuscript.

- Figure S2: “TAGLN(D)” => “TAGLN (D)”

- Table S1: Specify “Forward” and “Reverse” considering “(5’-3’)” like in Table S2.

Comments on the Quality of English Language

Well written, no further comments.

Reviewer 2 Report

Comments and Suggestions for Authors

The study by Zhang et al.  investigated the effect of the CoQ10 on the phenotypic transformation of the FBN1 knock-down smooth muscle cells. The authors suggest that FBN1 knock-down in the primary human SMCs could be used as a cell model of Marfan Syndrome.  The knock-down of FBN1 cells was accompanied by a decreased TFAM expression and mitochondrial mass, and an increase in inflammatory cytokines. The authors also found that Coenzyme Q10 treatment restored mitochondrial function of the cells and reversed the phenotypic transformation of shFBN1 SMCs to the synthetic phenotype.

The study indeed contributes to our knowledge about the role of mitochondrial function in the SMCs and their possible therapeutic regulation by aneurysms. Nevertheless, the changes in regulation of the main genes should be confirmed on the protein level.

 Comments:

1.    Reconsider the title: ” …..in a Marfan Syndrome Cell Model” , perhaps more precise would be “in SMCs with FBN1-Knockdown “.

2.    Regulation of gene expression should be confirmed by protein expression.

3.    Does Rotenone regulate TFAM level and oxidative phosphorylation in control cells? (Figure 3).

4.    The concentration of CoQ10 should be explained and indicated in the Figure Legends.

Round 2

Reviewer 2 Report

Comments and Suggestions for Authors

The authors have answered the questions, performed additional Western Blot analysis and improved the manuscript. 

Minor comment:

Please add the molecular weights of the proteins in Figure 6 C Western Blot.
